# Impact of Dietary Patterns on Plaque Acidogenicity and Dental Caries in Early Childhood: A Retrospective Analysis in Japan

**DOI:** 10.3390/ijerph19127245

**Published:** 2022-06-13

**Authors:** Yukie Nakai, Yukako Mori-Suzuki

**Affiliations:** 1Department of Dental Hygiene, University of Shizuoka Junior College, Shizuoka 422-8021, Japan; 2Suzuki Dental Clinic, Hiroshima 732-0822, Japan; m_yukkann0127@yahoo.co.jp

**Keywords:** cariogenic diet, dental caries, dietary sugars, feeding behavior, oral health

## Abstract

This study aimed to assess the relationship of dietary patterns, such as frequency, timing, and cariogenicity of food/beverage consumption, with plaque acidogenicity and early childhood caries (ECC) in Japan. A total of 118 children aged 1–4 years who had visited the pediatric dental clinic were enrolled. We retrospectively reviewed their records to collect data including age, sex, medical history, medication, caries status, and plaque acidogenicity level at the first dental visit. The plaque acidogenicity level was measured using Cariostat^®^. Dietary data were collected from 3-day dietary records, and the dietary cariogenicity score was calculated from these data. Children with ECC or high plaque acidogenicity consumed between-meal sugars more frequently than did their counterparts (*p* = 0.002 and *p* = 0.006, respectively). Children with ECC or high plaque acidogenicity drank juices between meals more frequently than at mealtimes (*p* = 0.02). Frequent consumption of between-meal sugars was associated with higher plaque acidogenicity and ECC, and frequent breast/bottle feeding was associated with ECC. No differences were found in the dietary cariogenicity scores between these groups. Therefore, the frequency and timing of sugar consumption, might affect plaque acidogenicity and ECC, and reducing the frequency of sugar intake could prevent ECC.

## 1. Introduction

Dental caries develops from a complex interaction among cariogenic bacteria, diets, and susceptible hosts. Diet is a major modifiable contributing factor to caries development. Frequently exposing teeth to fermentable carbohydrates, especially sugars, for extended periods of time generates acidic plaque environment, which promotes tooth enamel demineralization and causes dental caries [1]. Thus, diets rich in sugars increase caries development risk, and sugar restriction has been recognized as an important strategy for reducing it. A recent guideline by the World Health Organization (WHO) recommends reducing free sugar intake to below 10% of daily energy intake or even to below 5% to reduce health problems such as obesity and caries development [2]. Free sugars are defined as all monosaccharides and disaccharides added to foods by manufacturers, cooks, or consumers and sugars naturally present in honey, syrups, fruit juices, and fruit juice concentrates [2]. There are two possible dietary ways to restrict sugar intake for caries prevention: reducing the amount of added sugar to foods or drinks or reducing the frequency of sugar consumption. Determining the importance of reducing the total amount versus frequency of sugar consumption is challenging because these two factors are difficult to differentiate between: if either one increase, the other may uncontrollably increase. Some previous studies showed that the frequency of sugar consumption, rather than the amount, is related to caries [3,4]. Palmer et al. classified food cariogenicity and reported that the frequency of highly cariogenic food/beverage intake or between-meal juices was associated with early childhood caries (ECC) in 2-to 6-year-old children in the United States (US) [5]. Hong et al. reported that frequent consumption of foods and drinks with added sugar was associated with caries development among UK children aged 12–15 years [6]. Zeng et al. also found that the consumption frequency of sugary snacks and drinks and frequency of snacking before sleep were associated with caries development among Chinese children aged 3–5 years [7]. These studies provided scientific evidence for dental and medical professionals to focus their dietary advice on reducing frequency of sugar consumption in clinical settings. Frequency fits better in our theoretical understanding of the caries process. Therefore, frequency-based goals may be more practical for patients to follow than goals based on the amount of added sugars [8]. However, varied dietary preferences and habits among children in different countries with different cultural backgrounds may affect feeding behaviors and subsequently influence caries risk and occurrence of dental caries. Consequently, these previous findings need to be validated in other populations. Several studies have reported that highly cariogenic foods/beverages or certain scores that reflect dietary cariogenicity were associated with increased levels of cariogenic bacteria, such as mutans streptococci, or increased caries prevalence [5,9,10,11]. Many bacteria other than mutans streptococci in dental plaque can produce acids from sugary foods and drinks. These acids might demineralize tooth enamel. However, if the acidification episodes are mild and infrequent, homeostatic mechanisms in the plaque may easily restore the mineral balance in favor of remineralization [12]. Frequent acidification of plaque through poor dietary habits, such as frequent consumption of sugar-sweetened beverages and foods, increases the acidogenic/aciduric bacteria and subsequently leads to the dominance of mutans streptococci. Likewise, detection of lactobacilli and *Bifidobacterium* in ECC lesions [13,14] is in accordance with the extended ecological plaque hypothesis [12]: both bacterial genera are aciduric enough to colonize and proliferate in acidic carious lesions [15]. Acidogenicity of plaque microbiota (plaque acidogenicity) seems to be a suitable caries risk indicator that fits more in the model of the ecological plaque hypothesis [16] than do other factors. Although there are many studies on dental caries, no other study has ever evaluated associations between dietary patterns and plaque acidogenicity in children. We hypothesized that dietary patterns such as frequency, timing, and cariogenicity of food/beverage consumption affect plaque acidogenicity and distinguish children with ECC from caries-free children. Therefore, this study aimed to assess whether frequency of between-meal sugar consumption could influence caries risk and caries occurrence in Japanese children.

## 2. Materials and Methods

### 2.1. Participants and Procedures

This was an observational retrospective study. All data were retrospectively collected from the dental chart records of pediatric patients who had their first visit between September 2011 and November 2012 to Hello dental clinic, a part of the medical corporation of the Miyake Obstetrics and Gynecology Clinic, Okayama, Japan. All procedures were performed in accordance with the Declaration of Helsinki. The Ethics Committee of the OB-GYN Clinic Medical Corp. approved the study protocol (approval number: 2013-4). The parents of the pediatric patients were informed that their children’s data would be irreversibly de-identified and used for research with a chance to opt-out. This study is reported in accordance with the STROBE guidelines [17]. The following patient data were collected at the first dental visit of each patient: age, sex, medical history, medication, number of erupted teeth, caries status, and results of the caries activity test (Cariostat^®^ test; Dentsply Sirona KK, Tokyo, Japan). These data were retrieved from the charts by the pediatric dentist (Y.M.-S.), who was responsible for providing oral examinations/tests, and were entered into a computer database (Microsoft Excel). Dietary information was obtained from 3-day dietary records filled out by parents at baseline. The inclusion criteria were as follows: (i) children aged 1–4 years old at the time of initial examination; (ii) a complete clinical data set including oral examination findings, results of the Cariostat^®^ test [18,19,20], which is a caries activity test (showing acidogenicity levels in plaque samples), and dietary records; (iii) no previous experience of any dietary instructions for the purpose of caries control; and (iv) children with primary dentition. Children with systemic disorders or use of antimicrobials within the preceding month were excluded from the study. The sample size was estimated using the following assumptions previously reported [21]. If the power was set to 80% and the level of significance was < 0.05, the minimum sample size was 28 participants per group.

### 2.2. Outcome Variables

Presence of caries in primary teeth and plaque acidogenicity were the two dependent variables in this study. Clinical examination and plaque sampling for the Cariostat^®^ test were implemented by two well-trained and experienced pediatric dentists in a fully equipped dental chair or using the knee-to-knee position in cases of uncooperative children. Dental caries assessments were performed using an explorer and a mouth mirror under good illumination and with bitewing X-rays, when necessary. According to the WHO criteria [22], caries was characterized when a lesion had a detectable cavity and undermined enamel in a pit, fissure, or on a smooth surface. The number of decayed (cavitated), missing (due to caries), and filled primary teeth were counted to record the dental caries experience (dmft index). ECC was defined as the presence of one or more decayed, missing, or filled primary teeth in a child aged ≤ 71 months [23]. The children were dichotomized as ECC-positive (dmft ≥ 1) or caries-free (dmft = 0) based on their caries status. The Cariostat^®^ test [18,19,20] was routinely performed to measure the caries activity level in all children as part of comprehensive examination at their first visit to the dental clinic. Plaque samples were analyzed for microbe acidogenicity. The analysis procedures were as follows: one of two well-trained pediatric dental specialists used sterile cotton swabs to collect plaque samples from the buccal and labial surfaces of the patient’s maxillary teeth using a continuous side-to-side light sweeping stroke that was repeated a couple of times. Thereafter, these cotton swabs were immediately immersed into the Cariostat^®^ test medium. After incubating this medium at 37 °C for 48 h, colorimetric changes in the medium that reflected plaque acidogenicity were assessed by a well-trained dental hygienist, with standard color tubes provided by the manufacturer as references. The Cariostat scores “0: (blue),” “1: (green),” “2: (yellow-green),” and “3: (yellow)” corresponded to plaque pH values of 7.0, 5.5, 4.5, and 4.0, respectively. Values between 0 and 1, 1 and 2, and 2 and 3 were rounded to 0.5, 1.5, and 2.5, respectively. Scores of <2 and ≥2 indicated low and high levels of acidogenicity, respectively [18].

### 2.3. Dietary Data

Food and beverage consumption was analyzed using 3-day dietary records. Record sheets were provided to the participants’ parents with oral and written instructions at their first visit. The parents were asked to choose three different days (one weekday, in addition to both Saturday and Sunday) within the following week and begin recording the identity and time of any food or drink consumed each day, including breast/bottle feeds. They were also instructed not to change their routine eating habits during those three days and bring filled records with them at their second visit (within approximately two weeks). The completed records were intended to be examined and used for dietary consultations. 

### 2.4. Data Analysis

It was hypothesized that dietary factors would help differentiate children with ECC from caries-free children, as well as individuals with high levels of plaque acidogenicity from those with low levels. The dietary survey data included the counts per child regarding the number of items consumed in each of the five food categories (cario 00, caries-protective; cario 0, non-cariogenic; cario 1, low cariogenicity; cario 2, liquids; cario 3, solid-retentive food) based on their potential cariogenic risk [5,24], and the total number of food and drink items. To summarize dietary cariogenicity scores for each child, 3-day dietary records were scored by multiplying the frequency score for each item by a cariogenicity rating (5 points: 0–4, “caries-protective” to “highly cariogenic”) based on the Palmer cariogenicity classification [5,24], then summing the ratings to obtain an overall score. Human breast milk and infant formulas were included in the same category (“low cariogenicity”) as “milk” in the analysis. Food and beverage frequencies represented number of half-hour intervals; consumption was categorized as meals and between-meals based on consumption times and nature of foods. The counts of caries-free and ECC children or high and low levels of plaque acidogenicity, categorized by consumption frequency per 3 days, were evaluated by chi-square test for trends in proportion. The pattern of beverage intake (with meals or between meals), including that of milk, juice, water, and tea intake, was examined using two sample *t-tests*. Caries-free and ECC children, or those with high and low plaque acidogenicity, were compared by using *t-*tests. Logistic regression analyses were performed to evaluate the hypothesis that cariogenicity, frequency of between-meal sugar consumption, and feeding practices could influence plaque acidogenicity and caries development in this population. Each dietary frequency variable was dichotomized (between-meal sugar consumption: >6 times per 3 days vs. ≤6 times per 3 days, breast/bottle feeding: ≥8 times per 3 days vs. <8 times per 3 days). These variables were included because they are known to be associated with caries and have been commonly used in other studies [9]. Dietary cariogenicity scores were dichotomized such that “0” was designated for a cariogenicity score of <110 per 3 days and “1” for a score of ≥110, which is in the 75th percentile. Statistical significance was set at *p* < 0.05. Statistical analyses were performed using the IBM SPSS version 23 software (IBM Corp., Armonk, NY, USA).

## 3. Results

### 3.1. Clinical and Demographic Data

A total of 118 children (boys, 50%) who underwent oral examinations and plaque acidogenicity testing and had complete dietary records were included in this study. Of these, 30 (25.4%) had ECC (dmft ≥ 1). The mean dmft was 1.2 ± 2.7 (± standard deviation [SD]; range, 0–13). The Cariostat score distribution was 0% (n = 0, score = 0), 7.6% (n = 9, score = 0.5), 26.3% (n = 31, score = 1.0), 22.9% (n = 27, score = 1.5), 31.4% (n = 37, score = 2.0), 8.5% (n = 10, score = 2.5), and 3.4% (n = 4, score = 3.0) (due to rounding errors, the total was not 100.0%). Therefore, 43.2% of the children showed high plaque acidogenicity (score ≥ 2.0). Table 1 shows the characteristics of the participants. There was no difference between the number of boys and girls. The children with ECC or high plaque acidogenicity were significantly older and had more erupted teeth than their counterparts. The average age of the participants was 1.76 years (1.85 and 1.68 years for boys and girls, respectively). The average number of erupted teeth was 15.1 (14.9 and 15.3 for boys and girls, respectively). There were no significant differences in age, number of erupted teeth, caries status, or plaque acidogenicity levels between boys and girls. All participants had dietary records completed by their mothers only.

### 3.2. Food and Drink Frequencies

The mean total frequency of between-meal sugar consumption was 5.7 ± 3.2 times in 3 days (range, 0–18 times). The children with ECC consumed between-meal sugars significantly more frequently than caries-free children (7.6 ± 4.1 vs. 5.0 ± 2.6, *p* = 0.002) (Table 2). The children with high plaque acidogenicity also consumed between-meal sugars significantly more frequently than those with low acidogenicity (6.7 ± 3.8 vs. 4.9 ± 2.5, *p* = 0.006) (Table 2). The proportion of those with ECC increased with elevated frequency of between-meal sugars (*p* = 0.03) (Figure 1). The proportion of those with high plaque acidogenicity also increased with elevated frequency of between-meal sugars (*p* = 0.04) (Figure 2). Regarding drinks, the children with ECC consumed juices between meals significantly more frequently than at mealtime (2.70 ± 2.70 vs. 1.44 ± 1.70, *p* = 0.02) (Figure 3). Those with high plaque acidogenicity consumed juices between meals significantly more frequently than at mealtime (2.25 ± 2.36 vs. 1.39 ± 1.72, *p* = 0.02) (Figure 4). No significant differences were found for any of the other beverages. Feeding practices in this study population revealed that 24.6% (29/118) of the participants had been breastfed/bottle-fed with a mean frequency for 3 days (range: 1–24) of 7.6 ± 6.8. Breast/bottle feeding decreased with increasing age: 39.7% (25/63) in 1-year-olds, 10.3% (3/29) in 2-year-olds, 5.9% (1/17) in 3-year-olds, and 0% (0/9) in 4-year-olds.

### 3.3. Dietary Cariogenicity

Table 2 shows significant differences in the frequency of consumption of non-cariogenic food (cario 0) (11.2 ± 5.6 vs. 8.8 ± 4.5; *p* = 0.02) and liquids (cario 2) (6.1 ± 3.9 vs. 4.1 ± 3.6; *p* = 0.009) between the ECC and caries-free children. There were no significant differences in the consumption of foods categorized as caries-protective (cario 00), low cariogenic (cario 1), and solid-retentive food (cario 3) between those with and without ECC. No significant differences in the frequency of any food/drink categories were found between plaque acidogenicity levels. The mean dietary cariogenicity score was 95.4 ± 23.0 (range: 42–175). No differences were found in dietary cariogenicity score between these groups.

### 3.4. Logistic Regression

Logistic regression analyses were performed to evaluate the hypothesis; the adjusted odds ratios (ORs) and 95% confidence intervals (CIs) are shown in Table 3. The children who consumed between-meal sugars more frequently (>6 times/3 days) were 4.2 times significantly more likely to develop ECC (95% CI: 1.1–15.8; *p* = 0.03) and 3.9 times significantly more likely to show high plaque acidogenicity (95% CI: 1.3–11.1; *p* = 0.01) than their counterparts. The children who were breast/bottle-fed more frequently (≥8 times per 3 days) were 10.7 times significantly more likely to develop ECC (95% CI: 1.1–102.6; *p* = 0.04) than their counterparts. However, these breast/bottle-fed infants did not differ from their counterparts in plaque acidogenicity. The dietary cariogenicity score showed significant associations with neither plaque acidogenicity nor ECC.

## 4. Discussion

This retrospective study provided evidence of the relationship between dietary patterns and plaque acidogenicity and dental caries in early childhood in Japan. The total annual sugar consumption in the Japanese population has been exceptionally less compared to that in other developed countries (Japan, 16.8; United States, 33.3; European Union, 38.2; and Australia, 49.7 kg/capita/year in 2014–2016) [25]. Regarding the amount of consumed sugar, Japanese people consume less sugar than those in other countries. Accordingly, for caries prevention by restricting sugar consumption in Japanese patients, setting frequency-based goals makes more sense than goals based on the amount consumed. Dietary patterns regarding sugar consumption are considered to determine the oral environmental conditions for the resident plaque microbiota and significantly alter their natural balance according to the ecological plaque hypothesis [16]. Considering the ecological changes in the plaque, the indicator for caries risk needs to involve the total acidogenic and aciduric capacities of the spectrum of plaque bacteria rather than focusing only on the presence or amount of specific cariogenic bacteria such as mutans streptococci. Therefore, the Cariostat^®^ test measuring plaque acidogenicity was chosen for assessment of caries risk. To our knowledge, this study is the first to evaluate the impact of dietary patterns on ECC and plaque acidogenicity in an early childhood population with a background of non-Western culture.

The frequency of between-meal sugar consumption was shown to be associated with ECC and plaque acidogenicity in this study population. The findings regarding ECC were in accordance with the results of earlier studies in children in other countries with different cultures or socio-economical levels [5,6,7,26,27]. In this study, the frequency of between-meal sugar consumption was operationalized as >6 times in 3 days versus ≤6 times in 3 days during logistic regression analyses, which corresponds to >twice a day versus ≤twice a day. With this operationalization, food consumption >twice a day (or ≥3 times a day) was seen to impact ECC (OR: 4.2, 95%CI: 1.1–15.8) and plaque acidogenicity (OR: 3.9, 95%CI: 1.3–11.1). Based on our findings, we might emphasize that this frequency is an appropriate cutoff for recommendation. An earlier study among 9-year-olds in the Netherlands concluded that food consumption 7 times per day was an appropriate cutoff [28].

The association of between-meal juice intake with ECC, being compared with mealtime and other drinks, is consistent with the results in US children [5]. These findings confirmed that eating and drinking during meals might not be a major risk factor for caries development [5,26]. Intake of some specific foods/drinks that were categorized into non-cariogenic diet and liquid cariogenicity classifications were shown to be associated with ECC. However, the dietary cariogenicity score, which reflects both the cariogenicity and frequency of all food/drink items consumed for a total of 3 days, did not show significant associations with ECC or plaque acidogenicity in this study. This is inconsistent with the results of earlier studies that showed a significant relationship between such a cariogenicity score and ECC in US children [5,9].

This study validated the hypothesis that dietary factors, such as frequency and timing of sugary food/beverage consumption, were associated with plaque acidogenicity and ECC. This study included both breast/bottle use in the dietary information, thus providing an opportunity to assess breast/bottle feeding on plaque acidogenicity and ECC, which has rarely been performed before. The frequency of breast/bottle feeding did not contribute to plaque acidogenicity. However, this frequency contributed to ECC occurrence in this study. Previous studies using a de-salivated rat model reported that human milk was more cariogenic than cow milk [29,30]. This could be because human milk contains a higher concentration of lactose (7% vs. 3%), resulting in reductions in plaque pH level; less calcium (22.0 vs 114 mg) and phosphate (9.8 vs. 96 mg/100 g); and lower protein levels (1.2 vs. 3.3 g/100 mL) than in cow milk. However, human milk is known to be less cariogenic than sucrose or sugar [29]. The cariogenicity of infant formulas remains inconclusive because it depends on the carbohydrate content of each product [31,32]. A recent systematic review showed that bottle-fed children had significantly more ECC than breastfed children and indicated that breastfeeding could be protective against caries in early childhood [33]. A population-based cohort study conducted in Brazil reported that infants breastfed for ≥24 months increased their caries risk at 5 years of age [34]. Another critical review described a benefit of breastfeeding up to 12 months, with a positive association between caries and breastfeeding for a longer period [35]. The findings of this study confirmed that breastfeeding or bottle use over 1 year was positively associated with ECC in Japanese children. Feeding behaviors that result in prolonged and frequent exposure of acid to tooth surfaces in young infants with breast/bottle use, regardless of the relatively lower cariogenicity of human milk or formulas, may raise their cariogenic potential, thereby causing caries.

The use of dietary records is considered an effective and reasonably accurate method for measuring an individual’s food and beverage consumption [36]. Regarding dietary assessments in children, 3-day dietary records with parents as proxy reporters were thought to be the most accurate method [37]. However, there is still a possibility of misreporting intake, as is the case with any diet record. We recognized the limitations in obtaining thorough dietary records of a 3-day period; this is a short time and does not allow drawing of conclusions regarding the long-term impacts of this dietary pattern. However, previous studies with dietary surveys over an even shorter period, such as a 24-h diet recall of the previous typical day, have provided valuable findings [5,38,39]. We assumed that obtaining the most possible typical frequencies from the dietary records was important regardless of the period itself. Therefore, we excluded participants who had previously received dietary instructions. Although dietary records are a standard method for dietary assessment, records of a longer period may present a huge burden on participants. The open-ended dietary record used in this study allowed us to capture consumption frequencies on typical days compared with questionnaires that capture the consumption frequency of specific foods and beverages at daily and weekly intervals.

There are several limitations to this study. First, caries assessments were performed by two experienced dentists without inter-rater reliability calibration. However, diagnostic disagreement might be minimal because they had been calibrated in a previous study [40]. Furthermore, the data were used for actual dental treatments and checked for accuracy by another pediatric dentist. Second, the findings may not be generalizable to other populations because this sample was homogeneous. Diet varies based on culture, socio-economic status, or health consciousness; therefore, further investigation in other populations with distinct contextual factors is required to examine the generalizability of our findings. Third, this study population might have missed children with extremely high risks of caries, or extremely severe ECC, who would not have been brought to the dentist until manifestation of any symptoms. Fourth, no other behavioral factors related to toothbrushing and use of fluoride, which might act as confounders, were included in this analysis.

Nevertheless, the strength of this study might be the high response rate of nearly 100% (data not shown). Almost all patients had completed records at the second dental visit in the dental clinic. The 3-day dietary records of the participants were routinely obtained in the dental clinic because the dentist or dental hygienist explained its clinical purpose on a face-to-face basis individually until the parents accorded; this may have encouraged the high response rate. In conclusion, this study confirmed that 1–4–year–old Japanese children were more likely to experience dental caries and/or to have high plaque acidogenicity if sugar consumption between meals was frequent. Frequent breastfeeding was more likely to increase the risk of dental caries. A larger and prospective study should be conducted to further elucidate the impact of dietary patterns on plaque acidogenicity and ECC.

## 5. Conclusions

This study confirmed that 1–4–year–old Japanese children were more likely to experience dental caries and/or to have high plaque acidogenicity if they had frequent sugar consumption between meals. Frequent breastfeeding was more likely to increase the risk of dental caries. A larger and prospective study should be conducted to further elucidate the impact of dietary patterns on plaque acidogenicity and ECC.

## Figures and Tables

**Figure 1 ijerph-19-07245-f001:**
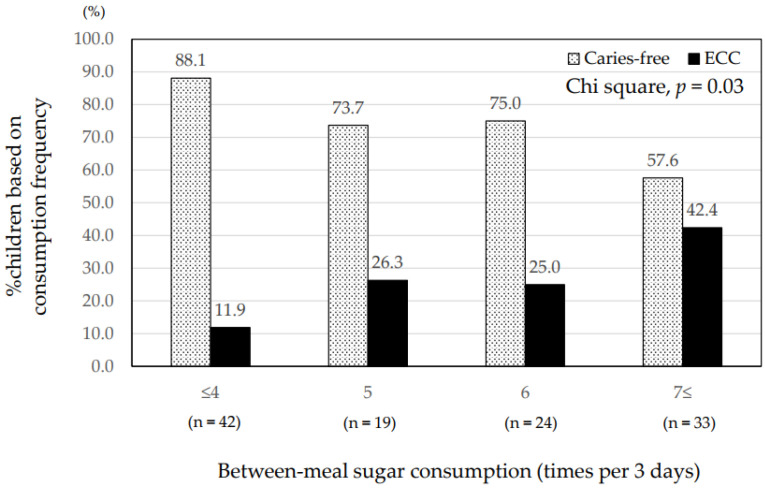
Frequencies of between-meal sugar consumption and caries status. The proportions of children with ECC and caries-free children differ depending on the frequency between-meal sugar consumption for 3 days, based on survey responses. The proportion of children developing ECC increases with elevated frequency of between-meal sugar consumption (*p* = 0.03; chi-square test). ECC, early childhood caries.

**Figure 2 ijerph-19-07245-f002:**
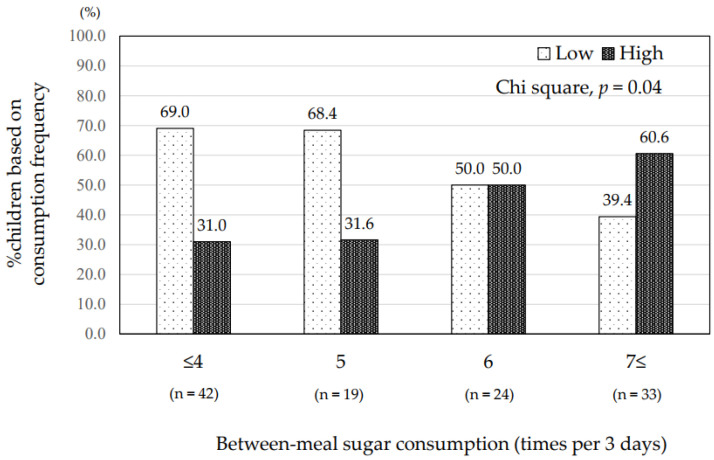
Frequencies of between-meal sugar consumption and plaque acidogenicity levels. The proportions of children with high and low levels of plaque acidogenicity differ depending on the frequency of between-meal sugar consumption for 3 days, based on survey responses. There is a significant increase in high plaque acidogenicity in children consuming sugar more frequently, with an opposite trend occurring in children with low acidogenicity (*p* = 0.04; chi-square test).

**Figure 3 ijerph-19-07245-f003:**
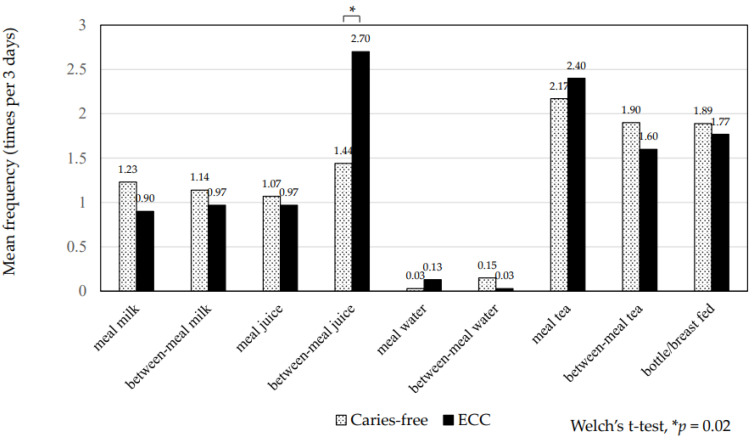
Frequencies of mealtime and between-meal beverages and caries status. The mean frequency of each beverage at mealtimes and between–meal beverages is shown. Children with ECC drink juices between meals significantly more frequently than caries-free children (*p* = 0.02; Welch’s *t*-test). ECC, early childhood caries.

**Figure 4 ijerph-19-07245-f004:**
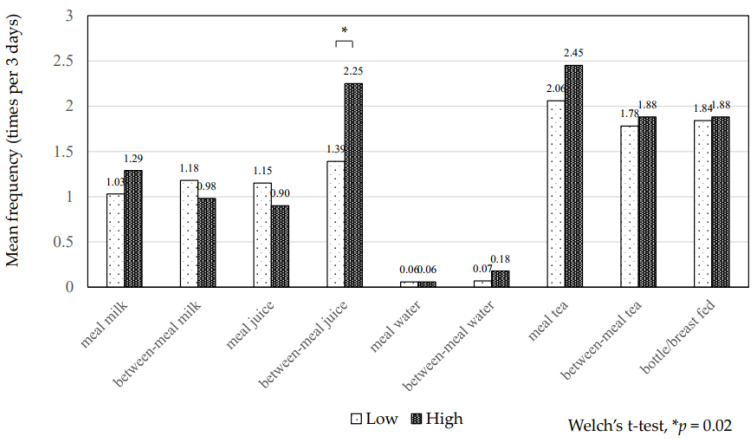
Frequencies of mealtime and between-meal beverages and plaque acidogenicity levels. The mean frequency of each beverage at mealtimes and between meals is shown. The children with high plaque acidogenicity drink juices significantly more frequently between meals than those with low acidogenicity (*p* = 0.02; Welch’s *t*-test).

**Table 1 ijerph-19-07245-t001:** Participant characteristics.

	Total Sample (n = 118)	Caries Status		*p-*Value	Plaque Acidogenicity		*p-*Value
		ECC (n = 30)	Caries-Free (n = 88)		High (n = 51)	Low (n = 67)	
	mean ± SD	mean ± SD			mean ± SD		
Age (years)	1.76 ± 0.97	2.7 ± 1.1	1.4 ± 0.7	**<0.001**	2.0 ± 1.0	1.6 ± 0.9	**0.01**
No. of erupted teeth	15.1 ± 5.1	18.1 ± 3.3	14.1 ± 5.2	**<0.001**	16.2 ± 4.4	14.2 ± 5.4	**0.03**
	n (%)	n (%)			n (%)		
Sex							
Girl	59 (50)	14 (46.7)	45 (51.1)	0.83	22 (43.1)	37 (55.2)	0.27
Boy	59 (50)	16 (53.3)	43 (48.9)		29 (56.9)	30 (44.8)	

Significant *p*-values are marked in bold. ECC, Early childhood caries; SD, Standard deviation.

**Table 2 ijerph-19-07245-t002:** Cariogenicity of food intake for 3 days

	Total Sample (n = 118)	Caries Status		*p*	Plaque Acidogenicity		*p*
		ECC (n = 30)	Caries-Free (n = 88)		High (n = 51)	Low (n = 67)	
	Mean ± SD	Mean ± SD			Mean ± SD		
Food intake for 3 days of survey							
Caries-protective (cario 00)	4.5 ± 4.6	4.2 ± 4.4	4.6 ± 4.6	0.65	4.6 ± 4.7	4.5 ± 4.5	0.83
Non-cariogenic (cario 0)	9.4 ± 4.9	11.2 ± 5.6	8.8 ± 4.5	**0.02**	9.4 ± 5.0	9.5 ± 4.9	0.92
Low cariogenic (cario 1)	19.5 ± 7.4	19.0 ± 7.2	19.6 ± 7.5	0.71	19.1 ± 7.5	19.8 ± 7.4	0.61
Liquids (cario 2)	4.6 ± 3.6	6.1 ± 3.9	4.1 ± 3.6	**0.009**	5.1 ± 3.7	4.2 ± 3.5	0.17
Solid/retentive food (cario 3)	8.3 ± 3.4	8.0 ± 4.1	8.4 ± 3.1	0.52	8.5 ± 4.0	8.2 ± 2.9	0.60
Total food and drink items consumed	46.3 ± 11.5	48.5 ± 12.8	45.6 ± 11.0	0.23	46.7 ± 11.9	46.0 ± 11.3	0.760
Dietary cariogenicity score ^a^	95.4 ± 23.0	99.4 ± 27.5	94.0 ± 21.2	0.27	96.9 ± 25.7	94.2 ± 20.8	0.540
Frequency of between-meal sugar consumption	5.7 ± 3.2	7.6 ± 4.1	5.0 ± 2.6	**0.002**	6.7 ± 3.8	4.9 ± 2.5	**0.006**
Frequency of breast/bottle fed	1.9 ± 4.7	1.8 ± 5.7	1.9 ± 4.3	0.904	1.9 ± 5.5	1.8 ± 3.9	0.957

^a^ Dietary cariogenicity score: 0 (cario00) +1 (cario0) +2 (cario1) +3 (cario2) +4 (cario3). ECC, early childhood caries; SD, standard deviation. The coding, cario 00 to cario 3, and the formula computing of dietary cariogenicity score are based on Palmer’s study [5]. A 3-day diet record was scored by multiplying the frequency for each item by a cariogenicity rating (5 points: 0–4, “carieo 00” to “cario 3,” respectively) based on the cariogenicity classification [5,24], followed by addition of the ratings to get an overall score. Significant *p*-values are bold-faced.

**Table 3 ijerph-19-07245-t003:** Logistic regression for factors associated with ECC incidence and high plaque acidogenicity.

Predictive Variables	Outcome Variables					
	ECC			High Plaque Acidogenicity		
	Adjusted OR	95% CI	*p-*Value	Adjusted OR	95% CI	*p-*Value
Frequency of between-meal sugar						
≤6 times per 3 days	1	reference		1	reference	
>6 times per 3 days	4.2	1.1–15.8	**0.03**	3.9	1.3–11.1	**0.01**
Frequency of breast/bottle fed						
<8 times per 3 days	1	reference		1	reference	
≥8 times per 3 days	10.7	1.1–102.6	**0.04**	3.40	0.6–20.4	0.18
Dietary cariogenicity score						
<110 per 3 days	1	reference		1	reference	
≥110 per 3 days	0.6	0.2–2.4	0.50	0.38	0.1–1.1	0.08

Adjusted for age in years, sex, and number of erupted teeth. ECC, Early childhood caries; CI, Confidence interval; OR; Odds ratio. Significant *p*-values are marked in bold.

## Data Availability

The data that support the findings of this study are available from the corresponding author upon request.

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
