# Peer review of "Impact of Dietary Patterns on Plaque Acidogenicity and Dental Caries in Early Childhood: A Retrospective Analysis in Japan"

_ijerph, 2022, doi:10.3390/ijerph19127245_

Round 1

Reviewer 1 Report

Dear Authors,

The article is interesting and valuable. Diet is main and modifiable agent influencing on caries development. Especially amount of sugar and frequent consumption of sugar daily has huge impact on dental plaque accumulation and oral health. Moreover, breastfeeding for a long period is an important factor of risk caries. I would like to underline, that the authors paid attention to this aspect in discussion and asked parents about breast feeding or bottle feeding. Therefore, conducted survey provided necessary  information of the side effects of prolonged breastfeeding and correlation between breastfeeding or bottle use over 1 year take into consideration age of studied patients.  

I am under impression of this article. The conducted study was very well planned, but should be carried out for longer time. In my opinion 3-day period time is not sufficient to evaluate  the impact diet on oral health. I didn’t find information about hygiene habits among children. Therefore, study should be better described. Authors should explain precisely why the decided take into consideration 3-day period in survey. It is main problem in conducted study.

The reviewer suggests major revisions.

Reviewer 2 Report

I would like congratulate the authors on their excellent work.

I believe that this article deals with a current and interesting topic and that it can be accepted for publication after some changes aimed at a greater understanding of the research carried out.

- Lines 46 and 49. References number 3 and 4 are quite dated as they are from 1954 and 1968. Please update them with most recent ones.

- Line 94. Sirona is misspelled

- Lines 117-118. This sentence needs a reference: "ECC was defined as the presence of one or more decayed, 117 missing, or filled teeth in any primary tooth in a child aged 71 months old or younger ".

- I would recommend adding in the Discussion section a part that explains advantages and limitations of the sample taken into consideration, compared to similar samples of patients from other geographical areas, from countries with a socio-cultural and economic level different from that of Japan.

Reviewer 3 Report

Thank you for your work. It’s a interesting study. The present study evaluated impact of dietary pattern on ECC and plaque acidogenicity in children in Japan. There are some points the authors should focus to: 

Introduction 

Objective – well elaborated

Please avoid to use the abbreviation for bacterium (e.g., MS) in the paper. 

Hypothesis – Pls clearly state the hypothesis

For using initial in the authors’ names in the text, it should be the first letter of the first name and the last letter of the last name.

Materials and Methods

For all materials and methodologies, please use general name/description in your text, followed by (brand name; manufacture, city, country).

Results

Figure 1 and 2 – the caption for (% of children) should be revise. For the stat differences means in all frequency, pls clearly clarify? In addition, pls clearly describe about  <4, 5, 6, 7< ? Also, do you have any specific reason for 6 times that was not differ with 5 times of sugar consumption between meal? Otherwise, should it be combined?  

For figure 3 and 4 – Pls clearly state about the type of meal and beverages in the material and methods. 

If you feel that your paper could benefit from English language polishing, you may wish to consider having your paper professionally edited for English language?

Round 2

Reviewer 1 Report

I accept the manuscript. Authors took into consideration all suggests and introduced explanation in text.